# Assessment of Protein Intake in the First Three Months after Sleeve Gastrectomy in Patients with Severe Obesity

**DOI:** 10.3390/nu13030771

**Published:** 2021-02-27

**Authors:** Lucrezia Bertoni, Romina Valentini, Alessandra Zattarin, Anna Belligoli, Silvia Bettini, Roberto Vettor, Mirto Foletto, Paolo Spinella, Luca Busetto

**Affiliations:** 1Department of Medicine, University of Padova, 35128 Padova, Italy; lucrezia.bertoni93@gmail.com (L.B.); romina.valentini@unipd.it (R.V.); alessandra.zattarin@aopd.veneto.it (A.Z.); anna.belligoli@gmail.com (A.B.); d.ssa.silvia.bettini@gmail.com (S.B.); roberto.vettor@unipd.it (R.V.); paolo.spinella@unipd.it (P.S.); 2Center for the Study and the Integrated Management of Obesity, Padova University Hospital, 35128 Padova, Italy; mirto.foletto@unipd.it; 3Clinica Medica 3, Azienda Ospedaliera di Padova, Via Giustiniani 2, 35128 Padova, Italy

**Keywords:** sleeve gastrectomy, protein intake, whey protein supplementation, weight loss

## Abstract

An adequate protein intake prevents the loss of fat-free mass during weight loss. Laparoscopic sleeve gastrectomy (SG) jeopardizes protein intake due to post-operative dietary restriction and intolerance to protein-rich foods. The purpose of this study is to evaluate protein intake in the first three months after SG. We evaluated, 1 month and 3 months after surgery, 47 consecutive patients treated with SG. Protein intake, both from foods and from protein supplementation, was assessed through a weekly dietary record. Patients consumed 30.0 ± 10.2 g of protein/day on average from foods in the first month, with a significant increase to 34.9 ± 4.8 g of protein/day in the third month (*p* = 0.003). The use of protein supplementation significantly increased total protein intake to 42.3 ± 15.9 g protein/day (*p* < 0.001) in the first month and to 39.6 ± 14.2 g of protein/day (*p* = 0.002) in the third one. Compliance with supplement consumption was 63.8% in the first month and only 21.3% in the third month. In conclusion, both one and three months after SG, protein intake from foods was not sufficient. The use of modular supplements seems to have a significant impact on protein intake, but nevertheless it remains lower than recommended.

## 1. Introduction

Bariatric surgery is the final alternative way, and the most effective solution, to achieve a sustained weight loss and to improve obesity-related pathological conditions in patients with severe obesity. Bariatric surgery involves surgical modifications of the gastrointestinal (GI) tract anatomy with a consequent alteration of nutrient flow affecting GI biology [1]. Sleeve gastrectomy (SG) and Roux-en-Y gastric bypass (RYGB), the most common bariatric procedures, engender weight loss not only by stomach restriction and nutrient malabsorption, but also by changing the secretion of a plethora of GI tract-derived signals playing a role in energy balance regulation [1]. It is fundamental for the patient to conceive bariatric surgery as a true change in life habits: an active involvement and consistent commitment is required to maximize the benefits and minimize the side effects after the procedure.

During a period of energy restriction, the macronutrient composition of the diet plays an important role in modulating the subsequent modification of body composition. Protein intake, in particular, has a role in modulating the loss of lean mass [2,3]. A moderately hyper-proteic dietary intake is indeed related to saving muscle mass, the preservation of the basal metabolic rate, and consequently the promotion of weight loss and maintenance [4,5]. Another aspect to consider is the proteins’ quality and biological value, determined by their amino acid composition and by their capability to proceed to synthesis. The muscular protein turnover is influenced by the type of amino acids (AAs) that are consumed. Essential AAs cannot be synthesized de novo by the body and must be introduced through the diet. Among these, branched-chain amino acids (BCAAs), leucine in particular, are characterized by the greater effect of the stimulation of protein synthesis [6].

Taking into account these considerations, recent nutritional post-bariatric guidelines and recommendations suggest the adoption of a relatively high protein intake in the first months after surgery, when the magnitude of energy restriction is larger and the risk for lean mass loss higher [7,8,9,10]. In particular, according to the 2019 Update of the Guidelines published by AACE/TOS/ASMBS, protein intake should be individualized, assessed, and guided by a registered dietitian, regarding gender, age, and weight. A minimal protein intake of 60 g/day and up to 1.5 g/kg ideal body weight per day should be adequate; higher amounts of protein intake (up to 2.1 g/kg ideal body weight per day) need to be assessed on an individualized basis [11]. The use of modular food supplementation based on whey proteins is deemed necessary until the patient’s natural diet reaches the established nutritional targets [11,12,13]. However, the implementation and applicability of these guidelines into the real world remain unclear, because only a few studies have tried to evaluate specifically actual protein intake in the first months after surgery. 

The aim of this study is to evaluate protein intake in the first post-surgery phase, i.e., 1 month and 3 months after laparoscopic SG. We took into consideration the protein intake related to natural foods and the eventual compliance to the prescribed modular protein supplement. Our analysis concerned protein quality, such as the detection of grams and percentages, of total proteins of animal and vegetable origin. The essential BCAA (leucine, isoleucine and valine) intakes were also evaluated.

## 2. Materials and Methods

### 2.1. Patients Enrollment and Clinical Pathway

For this study, 47 consecutive patients (33 women and 14 men) treated with SG at the “Centre for the Study and the Integrated Management of Obesity” of the Padova University Hospital were enrolled. Recruitment of the patients started in February 2016 and ended at the end of September 2016. The study was observational and did not require any treatment or potentially risky examination outside routine clinical care. All patients signed an informed consent form regarding the use of their anonymized clinical data for research purposes. The selection and preparation of the patients for SG was performed according to the clinical pathway adopted at our Institution for the diagnostic work-up and management of patients with obesity. 

Indication to bariatric surgery was based on international recommendations [14]. Briefly, surgery is recommended for patients with a BMI higher than 40 kg/m^2^ without other comorbidities or BMI higher than 35 kg/m^2^ with the typical obesity-related comorbidities. Alcohol addiction and severe psychiatric disorders (bipolar disorder, schizophrenia, bulimia, psychosis) are considered absolute exclusion criteria. Anxiety disorder, depression, binge eating disorders and night eating syndrome are not considered absolute exclusion criteria. Patients with these disorders are allowed to undergo surgery after a period of preliminary psychological or psychiatric therapy.

SG is performed as the first-choice procedure in our center, except for with patients affected by severe gastro-esophageal reflux disease. The surgical technique has been previously described [15]. Briefly, the procedure involved stomach longitudinal resection starting 4–5 cm from the pylorus with the preservation of the gastric antrum. SG was calibrated with a 34-Fr gastric bougie. As a 5–10% loss of body weight before surgery (1–2 months before) is considered useful in preventing surgical complications [16,17], all patients at our institution received a 4-week very low calorie diet (VLCD) of 800 kcal/day immediately before surgery.

After surgery, patients are required to follow a specific dietary protocol, aimed at compensating possible nutritional deficiencies. Patients were instructed by a registered dietitian as to dietary and behavioral norms, structured in such a way as to support the physiological adaptation of the reduced stomach. The diet, in the 7–10 days immediately after the surgery, must be liquid and composed of foods that do not need chewing: low-fat yogurt, fruit juices, pureed food, soup. It then becomes semi-liquid in the next 3–4 weeks, with the introduction of small-size pasta as a carbohydrate source and, if tolerated, protein foods (meat, fish, eggs) provided they are well cooked, minced and diluted in liquids, and then semi-solid for the next 3–4 weeks. When a certain level of adaptation is achieved, it is possible to adopt a solid diet of maintenance, that is more balanced and palatable. Behavioral norms to be adopted during the meal are also important: slow and accurate chewing, never drink during the meal but only up to 30 min before and at 30 min after, do not swallow excessive food volumes, avoid carbonated drinks, eat until satiation, avoid foods containing substances that could give burning or acidity, introduce foods rich in fibers gradually, start the meal with the protein dish. A multivitamin and mineral supplementation was prescribed to all patients according to current recommendations [13]. Up to the third month after the surgery, the use of whey protein integration is indicated in all patients, regardless of what is expected or observed to be the intake from foods. The defined amount is 30 g, to be taken during the day, diluted in sweet or savory, cold or lukewarm liquids.

### 2.2. Anthropometric Measurements

All anthropometric measurements were taken with the subjects wearing only light clothes without shoes. Height was measured to the nearest 0.01 m using a wall-mounted stadiometer. Body weight was determined to the nearest 0.05 kg using a calibrated balance beam scale. BMI was calculated as weight (kg) divided by the height squared (m^2^). 

### 2.3. Collection and Analysis of Dietary Records

At discharge, patients received detailed oral and written instruction on how to collect a weekly dietary record, requiring full registration of all foods and drinks consumed each day, including eventual protein supplements. The indication was to record the type and quantity of any modular and non-modular product taken, so that the effective protein intake could be calculated. The diary would cover the week preceding the first post-operative follow-up visit that is scheduled one month after surgery. At this visit, the weekly diary was collected by a dedicated dietitian and reviewed with the patients. Patients then received oral and written instruction on how to collect a weekly dietary record and were asked to fill the record during the week preceding the second post-operative follow-up visit scheduled three months after surgery. The diary was then collected and reviewed by the same dietitian at this visit.

The weekly dietary records were converted into grams of intake for any specific foods and the daily data were averaged along the week obtaining a mean daily data. Data were then entered and analyzed by using the software Winfood version Pro (Medimatica S.u.r.l., Colonnella, Italy). The software calculated total daily energy intake (kcal) and daily intakes of macro- and micro-nutrients (g) by using a very large database containing the compositions of over 1600 foods and supplements. The database is based on the alimentary tables released by the Italian National Institute of Research on Foods and Nutrition (Istituto Nazionale di Ricerca per gli Alimenti e la Nutrizione—INRAN). The following variables were considered for statistical analysis: (1)Composition of the diet at 1 month and 3 months after surgery;(2)Protein and BCAAs intake at 1 month and 3 months after surgery, considering the amount with and without protein supplement;(3)Overall protein intake in patients who have or have not taken the protein supplement;(4)Number and percentage of patients meeting the minimal recommended protein intake of 60 g/day [10] at 1 month and 3 months after surgery, considering the amount with and without protein supplement.

### 2.4. Physical Activity

During the first three months after surgery, patients were counselled to practice constant and regular low-intensity physical activity, while starting from the third month post-intervention the indication is to undertake a programmed physical activity that includes resistance and strengthening exercises. Both at 1 month and 3 months after the surgery, patients were asked to report the type of physical activity done daily, and to report the minutes. Minutes of physical activity usually practiced during the day at 1 month and 3 months after the surgery are registered.

### 2.5. Statistical Analysis

The statistical data analysis was performed using the software Statistical Package for the Social Sciences (SPSS, v.23.0) (IBM Corp., Armonk, NY, USA). Data were expressed as mean ± standard deviation (SD). Comparisons between data collected at month 1 and data collected at month 3 were performed by paired sample *t*-test. Within the individual observation times, comparisons were made between the data detected without and with the inclusion of the protein quota linked to the assumption of modular integration (paired sample *t*-test), and between patients who had or had not actually taken modular integration (Student’s *t*-test for unpaired data). In all analysis, a *p* value < 0.05 was considered to be statistically significant.

## 3. Results

The mean age of the 47 patients (33 women and 14 men) recruited in this study was 47 years, with a broad range (24–69 years). No patients were lost from the follow-up during the study. The anthropometric characteristics of the patients in the preoperative stage and the subsequent evolution at 1 month and 3 months after surgery are reported in Table 1. Weight loss was significant both 1 month and 3 months after surgery. 

The composition in grams of the usual dietary intake, without taking into consideration the additional intakes linked to modular protein supplementation, at 1 month and 3 months after the intervention is reported in Table 2. Total daily caloric intake, protein intake, and the intake of BCAAs increased significantly from 1 to 3 months after surgery. However, the mean protein intake was lower than the recommended daily protein intake of 60 g/day both at 1 month and at 3 months after surgery. 

The intakes of proteins and BCAAs observed at 1 month and 3 months after the surgery, without taking into consideration the intakes linked to modular protein supplementation and with the inclusion of the intakes linked to supplementation, can be compared in Table 3. At both evaluations, the use of the modular integration allowed a significant increase in the intakes of total proteins, animal proteins and BCAAs. However, the mean protein intake was lower than the recommended daily protein intake of 60 g/day both at 1 month and at 3 months after surgery, even taking into consideration the additional intakes from modular supplementation. 

The efficacy of modular supplementation in improving post-operative protein intake can be undermined by poor compliance. Indeed, according to the dietary records, only 63.8% of the patients regularly used modular supplementation at month 1 and 21.3% at month 3. A comparison of the overall protein intake in patients who have or have not actually taken protein supplementation at 1 month and 3 months is shown in Table 4. At month 1, the intakes of total proteins, animal proteins and BCAAs were significantly higher in patients who regularly consumed the protein supplementation. No significant differences were observed at 3 months, possibly in relation to the poor level of compliance. Again, in no cases was mean protein intake in line with the recommended daily protein intake of 60 g/day. 

Finally, we analyzed the number and percentage of patients meeting the minimal recommended protein intake of 60 g/day at 1 month and 3 months after surgery, considering the amount with and without protein supplement. Considering only intakes from foods, no patients met the minimal recommended protein intake at both 1 and 3 months. With the inclusion of the protein intakes from modular supplementation, only four patients (8.5%) met the minimal recommended protein intake at both 1 and 3 months. 

As for physical activity level, only 8 patients out of 47 started to practice regular physical activity in the last period (third month). Specifically, four patients attended a gym for 2 days a week, three patients went swimming or to an aqua gym twice a week, and one patient practiced dancing once a week. However, the weekly average of physical activity in minutes was 23.3 ± 16.9 in the first month and 23.6 ± 22.6 in the third month, without a significant change.

## 4. Discussion

Our aim was to evaluate to what extent the dietary restrictions, physiological adaptation over time and post-prandial side effects exerted an influence on food intake in the first three months after SG, with a particular emphasis on protein intake. We found that the protein intake from foods was very low during this period, and well below the recommended minimal requirements. The use of protein supplementation provides a significance increase in protein intake, but this effect is undermined by a general poor compliance with the supplementation. Furthermore, even in patients regularly assuming the supplementation, total protein intake remains lower than the recommended targets.

Loss of fat-free mass (FFM) and muscle mass is an unintended but quite unescapable consequence of energy restriction and voluntary weight loss. During medically induced weight loss, about 20–25% of weight loss is represented by FFM [8]. This proportion could be further increased in cases of very rapid weight loss, in particular during the first few months after a bariatric procedure [8]. Moreover, the first post-surgical phase could be characterized by an intensified condition of protein catabolism, inevitably induced by the surgical stress. Several studies indeed confirmed a significant loss of lean body mass during the first months after surgery [18,19,20]. Protein intake could have a role in modulating the loss of FFM [2,3], and a moderately hyper-proteic diet is associated with FFM preservation [4,5]. In particular, adequate protein intake in the first six months after SG was found to be associated with FFM saving [18]. Therefore, current guidelines [11,12,13] recommend the adoption of a relatively high protein intake in the first months after surgery, with a recommended protein intake ranging from 1.0 to 1.5 g/kg of ideal body weight, and with some patients requiring even higher amounts (2.1 g/kg of ideal body weight) [11]. However, only a few studies have so far examined the protein intake that diet could provide to a patient treated with bariatric surgery, and they found very low intakes [21,22]. In this study, we confirmed that protein intake from natural foods is well below the recommended levels in the large majority of patients in the first three months after SG. This was accompanied by a low intake of the essential BCAAs, leucine in particular, that are considered particularly important for protein synthesis [6]. 

The main reason for not achieving adequate protein intake after bariatric surgery is the difficulty of consuming solid protein-based aliments [23]. After SG, the consumption of solid protein-based foods (meat in particular) requires a careful and slow mastication, and these foods can more frequently elicit nausea, vomit, and regurgitation of food. Finally, proteins are considered the macronutrient which satiates the most [24,25]. 

Taking into account all these considerations, the current guidelines support the integration of the usual diet with an easy digestible liquid protein supplementation [11,12,13]. Modular supplementation based on whey proteins with a high BCAA content is usually preferred. The protein supplement (30 g/day) can be either consumed at different moments over the day or all at once during the main meal. Several randomized studies focused on how modular protein supplementation influences weight reduction and body composition after bariatric surgery, highlighting its positive impact [26,27]. In our observational study, we confirmed the efficacy of modular protein supplementation in improving post-operative protein and BCAA intake, but we also revealed that its effect can be undermined by a poor compliance, with less than two thirds of patients regularly taking the supplementation in the first month and less than a quarter in the third month. Several factors can affect the compliance to protein supplementation. First, patients frequently reported intolerance to the product, perceived as scarcely attractive and indigestible. Second, patients tend to underestimate the nutritional value of the product and its effective utility in preserving the lean body mass. Third, another element that could undermine compliance is the socioeconomic factors, as specific products are indeed very expensive. Cheaper alternatives can be found on the market, which however do not guarantee the sustainability of the products, as sometimes they do not match the required composition (i.e., non-modular protein supplements, composed of various macronutrients). Finally, the poor compliance to protein supplementation, and to dietary prescription at all, could be influenced by personality traits before surgery and to psychological factors in the post-operative period. Future research may detect whether some psychological factors may contribute to poor after-surgery dietary compliance.

The major limitations of our study are the small sample and the relatively short follow-up. We concentrated our attention on the first period after surgery, when the problem of protein intake is more important and the weight loss is more rapid. We also failed to collect reliable data on body composition changes, and therefore we cannot estimate to what extent the low protein intake observed in this study could really affect lean body mass preservation. Further studies with a more extended follow-up and the accrual of body composition data could be useful to further evaluate the clinical impact of our results. We also lack dietary data from the pre-operative period, but we consider it improbable that pre-operative dietary habits could have important influences on the dietary intake in the first post-operative periods, when nutritional quality and quantity are certainly more affected by the new GI anatomy. The role of potential confounders, like age, sex and physical activity levels, was not taken into account. However, our sample was mostly of female gender and limited to adult patients, and the recorded physical activity levels in the first three months after surgery were very low. Objective data about the reasons for lack of compliance, with recommendations regarding diet and physical activity and the use of protein supplements, were not recorded and they could only be derived from the available literature and our clinical impression. Finally, we have only evaluated patients treated with SG, the most performed surgery in our clinical setting, and we could assume that post-operative protein intake may be different after different types of bariatric procedure. Future research may investigate differences in protein intake in different types of bariatric treatments.

## 5. Conclusions

In conclusion, our study highlights the difficulties encountered in implementing nutritional recommendations regarding protein intake in the first months after bariatric surgery. Despite careful education and guidance by registered dietitians, the amount of protein derived from natural nutrition is not sufficient to reach the minimum required intake. As per guidelines, the use of protein supplementation is also recommended. The consumption of modular supplements of whey proteins seems to have a positive impact on the total protein intake. However, the compliance with this supplementation is poor, and the effect of protein supplements appears to be insufficient, even in compliant patients. 

## Figures and Tables

**Table 1 nutrients-13-00771-t001:** Anthropometric characteristics before and at 1 and 3 months after SG.

	Before (*n* = 47 Patients)	1 Month (*n* = 47 Patients)	3 Month (*n* = 47 Patients)
Weight, kg	121.2 ± 17.3	106.7 ± 15.1 ***	94.8 ± 14.3 ***
BMI, kg/m^2^	43.8 ± 5.4	38.6 ± 5.0 ***	34.3 ± 4.8 ***
EWL%	-	25.1 ± 8.3	45.0 ± 12.7

SG: sleeve gastrectomy. BMI: body mass index. EWL%: Percent of excess weight loss. Excess weight was calculated with a reference to the body weight corresponding to a BMI of 25 kg/m^2^. A paired sample *t*-test was performed between data before and the two observations after surgery. *** = *p* < 0.001.

**Table 2 nutrients-13-00771-t002:** Composition of the natural diet at 1 month and 3 months after SG.

	1 Month (*n* = 47 Patients)	3 Month (*n* = 47 Patients)	*p*
Energy, kcal/day	624.8 ± 188.0	722.2 ± 212.0	<0.001
Proteins, g/day	30.0 ± 10.2	34.9 ± 4.8	0.003
Animal proteins, g/day	13.5 ± 9.8	15.5 ± 7.9	0.127
Vegetal proteins, g/day	4.8 ± 5.8	5.2 ± 2.5	0.707
Leucine, mg/day	1362.6 ± 740.0	1663.5 ± 675.0	0.011
Isoleucine, mg/day	777.7 ± 426.0	936.1 ± 376.0	0.013
Valine, mg/day	923.7 ± 492.2	1106.4 ± 434.2	0.014
Fats, g/day	20.4 ± 8.4	28.3 ± 9.5	<0.001
Carbohydrates, g/day	84.0 ± 26.8	98.8 ± 29.8	0.001
Starch, g/day	29.6 ± 14.0	40.1 ± 17.3	0.001
Oligosaccharides, g/day	31.5 ± 14.3	35.4 ± 18.5	0.179
Fibers, g/day	6.4 ± 2.9	6.3 ± 2.6	0.879
Water, mL/day	1135.6 ± 332.6	1297.8 ± 327.1	0.003

SG: sleeve gastrectomy. Paired *t*-test 1 month vs. 3 months.

**Table 3 nutrients-13-00771-t003:** Comparison between protein and branched-chain amino acids (BCAAs) intakes from foods and with the addition of the amounts linked to modular integration at 1 month and 3 months after SG.

	Without Integration	With Integration	*p*
Month 1 (*n* = 47 patients)			
Proteins, g/day	30.0 ± 10.2	42.3 ± 15.9	<0.001
Animal proteins, g/day	13.5 ± 9.9	23.2 ± 15.2	<0.001
Vegetal proteins, g/day	4.8 ± 5.8	5.9 ± 7.2	0.159
Leucine, mg/day	1362.6 ± 740.0	2677.3 ± 1528.2	<0.001
Isoleucine, mg/day	777.7 ± 426.0	1590.7 ± 938.3	<0.001
Valine, mg/day	923.7 ± 492.2	1106.4 ± 434.2	0.014
Month 3 (*n* = 47 patients)			
Proteins, g/day	34.9 ± 9.6	39.6 ± 14.2	0.002
Animal proteins, g/day	15.6 ± 7.9	20.0 ± 13.5	<0.001
Vegetal proteins, g/day	5.2 ± 2.5	5.2 ± 2.5	0.183
Leucine, mg/day	1663.6 ± 675.9	2197.3 ± 14.2	<0.001
Isoleucine, mg/day	936.1 ± 376.9	1255.1 ± 809.7	<0.001
Valine, mg/day	1106.4 ± 434.2	1391.5 ± 5.0	<0.001

BCAAs: Branched-chain amino acids. SG: sleeve gastrectomy. Paired Student’s *t*-test for the intakes without vs. with the integration.

**Table 4 nutrients-13-00771-t004:** Comparison between overall protein intake in patients who have or have not actually taken protein supplementation at 1 month and 3 months after SG.

	Patients Not Compliant	Patients Compliant	*p*
Month 1 (*n* = 47 patients)			
Proteins, g/day	28.9 ± 9.9	49.8 ± 13.5	<0.001
Animal proteins, g/day	14.9 ± 12.4	27.9 ± 14.8	0.04
Vegetal proteins, g/day	6.0 ± 9.3	5.8 ± 5.8	0.938
Leucine, mg/day	1390.2 ± 694.9	3406.6 ± 1384.5	<0.001
Isoleucine, mg/day	781.8 ± 394.5	2049.0 ± 842.4	<0.001
Valine, mg/day	934.4 ± 462.2	2011.3 ± 827.5	<0.001
Month 3 (*n* = 47 patients)			
Proteins, g/day	34.7 ± 9.5	35.5 ± 10.4	0.837
Animal proteins, g/day	15.5 ± 7.8	15.8 ± 9.0	0.912
Vegetal proteins, g/day	5.2 ± 2.5	5.2 ± 2.6	0.958
Leucine, mg/day	1623.3 ± 621.7	1812.5 ± 870.6	0.532
Isoleucine, mg/day	921.6 ± 349.4	1003.9 ± 481.6	0.622
Valine, mg/day	1082.9 ± 403.8	1193.3 ± 548.2	0.564

SG: sleeve gastrectomy. Student’s *t*-test for unpaired data for the intakes in patients compliant vs. not compliant with the integration.

## Data Availability

The data presented in this study are available on request from the corresponding author.

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
