# Peer review of "Assessment of Protein Intake in the First Three Months after Sleeve Gastrectomy in Patients with Severe Obesity"

_nutrients, 2021, doi:10.3390/nu13030771_

Round 1
Reviewer 1 Report
The purpose of this observational study is to evaluate composition of the “natural” diet, protein intake, BCAA, the number and % of pts meeting the minimal recommended protein of 60 gm per day and recommended physical activity at the 1st and 3rd post-operative month after SG.
- Define “first period” after LSG and “natural”
- The title of your paper is “Qualitative and quantitative assessment of protein intake in the first period after sleeve gastrectomy in patients with severe obesity.” Using a qualitative assessment of protein intake is an original approach and could provide new information to this area, but it is not addressed in your paper. There is no mention or description of a qualitative data analysis nor any results. Please change your title to reflect the content of your study or include the qualitative methods, analysis, and results.
- The data is presented with associated statistical significance; however, there is no clinically meaningful discussion about the reported individual BCAA.
- There is no information provided from the Registered Dietitian’s collection of dietary records as to why these patients didn’t comply with recommendations such as using the protein supplement or being physically active.
- What reasons did the pts give for not taking protein?
- Cost, taste, other reason?
- Education – were the pts confused? Didn’t know which protein or activity?
- Education – did pts understand the importance of consuming the rec amount of protein or doing the activity?
- How were the instructions presented to pts?
- There are many unanswered questions here. This is where a qualitative data collection and analysis could have been useful and have provided new & interesting information.
- What reasons did the pts give for not taking protein?
- I think that this information alone, does not provide much benefit to the existing published research. I do think that a comparison of your findings (30 gm & 40 gm protein at 1 and 3 months) is very different from what has been reported in the U.S and could be interesting. But you would need to provide some answers as to why the protein intake is so low. You have reported that the protein intake is low due to poor patient compliance; but compliance is not well operationalized or measured.
Corrections:
- Line 16 last word of sentence change form to from
- Lines 47-51: Why did you delete “assessed, and guided by an RD” from the statement below?
“In particular, according to the 2013 Update of the Guidelines published by AACE/TOS/ASMBS, post-operative protein needs should be individualized assessed, and guided by an RD and corrected according to gender, age and weight, with a minimum protein intake set at 60 gr/day.
Mechanick et al., 2013 states: “Protein intake should be individualized, assessed, and guided by an RD, in reference to gender, age, and weight.
- Lines 47-51: Please provide further references for 120 gms protein s/p sleeve gastrectomy or delete “120 gms”
In your document: These recommendations correspond to a protein intake typically ranging from 1.0 to 1.5 gr/kg of ideal body weight, with some patients requiring even higher amounts (2.1 f/kg of ideal body weight) [10]. Therefore, the average protein requirement in the patient who underwent a sleeve gastrectomy is 60-120 gr/day or 1-1.5 gr/kg of ideal body weight, with individualization within the defined range.
- The defined range for s/p SG protein is not 120 gms per day.
Reference # 10 (Mechanick et al., 2013) does not support the recommendation of sleeve gastrectomy up to 120 gm protein per day.
In fact the defined range is… “(80–90 g/d) are associated with reduced loss of lean body mass” Mechanick, et al., 2013.
- Mechanick et al., 2013 is NOT the most recent guideline. Please use Mechanick et al., 2020 R38.
- Line 235 “semester” is used in academia but seems arbitrary here; please be more specific or use more conventional language.
- Lines 249-252 don’t make sense. I don’t understand what you are trying to say:
- Language in parts of the article is difficult to interpret, possibly have someone proficient in English review the article:
“Furthermore, the functional disturbs which normally verify shortly side effects which commonly occur after gastric restriction, such as vomiting, nausea, dysphagia and intolerance in the assumption of solid aliments, can may represent confounding factors instead of merely poor compliance an additional obstacle affecting total food intake, and protein intake especially.”
Finally, I believe that a journal with a focus on bariatric surgery might be more appropriate.
Reviewer 2 Report
This is an extremely useful paper. It is a small sample size and the authors acknowledge this.
As someone who deals with bariatric patients I am conscious that a number do not comply with presurgery advice which has implications for their health long term.
I very much valued this paper as it confirms the difficulties of adequate protein intake post-surgery and thus a need for more discussion and promotion of higher protein intake post-surgery and the promotion of foods with a higher level of protein plus appropriate supplementation.
The paper is well written and the methodology described. It is useful both practically and also the study could be replicated with a larger number of subjects
Reviewer 3 Report
In their article “ Qualitative and quantitative assessment of protein intake in the first period after sleeve gastrectomy in patients with severe obesity”, the authors evaluated the protein intake in the first 3 months (month 1 and month 3) after laparoscopic sleeve gastrectomy in 47 patients and observed that protein intake from natural nutrition was not sufficient. Their findings also suggested that consumption of nutrient supplements had a significant impact on protein intake which, however, remained lower than recommended. The manuscript is well-written and easy to follow. However, I have the following concerns:
- The authors may introduce what bariatric surgery is, so that it will be helpful for all the readers to understand.
- Line 41: It will be branched-chain amino acids.
- The usually accepted abbreviation for gram is “g” (not “gr”). Please check throughout the manuscript.
- Line 59: It will be “1 month” (I think “1” is missing here).
- Line 63: The essential BCAAs were also evaluated. But it is not detailed in the methods. It will be great if the authors may kindly explain how this was evaluated in the methods section.
- Line 67: The authors mentioned 47 patients but did not mention how many males and females were in the group. This has been mentioned later in the result section. It will be better if the authors consider mentioning that in the method section here.
- Line 80: “Anxiety disorder, depression, binge eating disorder and night eating syndrome are considered relative exclusion criteria and therefore they are subjected to a specific evaluation, with a possible use of a preliminary psychological or psychiatric therapy.” It is not clear what is meant by “relative exclusion criteria” and what “specific evaluation” was performed to exclude these conditions, since they may act as confounding factors for subsequent analysis.
- It is not clear how the results did not match as per recommendations. In line 93, it has been stated that “patients were instructed by a dietician to dietary and behavioral norms..” and in line 106, it has been mentioned “multivitamin and mineral supplements were prescribed…” but the results conclude that they were not sufficient (eg. Line 183, 199, 220, 241). This seems to be contradictory to each other and hence it will be appreciated if the authors kindly explain this discrepancy.
- In table 2: How was the energy calculated? Was it based solely on the amount of protein intake? If not, what other nutrient intake was considered, and how were they evaluated? The authors may consider describing the details in the method section.
- In table 2: Were the units per day? Please mention (eg. Kcal/day, g/day, etc.)
- The authors may consider restructuring the conclusion section. It is rather inconclusive to state “In conclusion, our study confirms that the problem of low protein intake after LSG 295 remained unsolved. The amount of protein assumed from natural nutrition is not sufficient to reach the minimum required intake.” Please consider elaborating these points and making the conclusion more specific.
- Did the authors mean “consumed” and “consumption” by the words “assumed” and “assumption” in the abstract? Please check.
- The article will benefit from grammatical and typographical corrections.
Reviewer 4 Report
The paper by Bertoni et al. evaluated protein intake in the first period after LSG.
The title and abstract are appropriate for the content of the manuscript. Globally, it is well conducted. Nevertheless, the study lacks some additional analysis that authors acknowledge as limitations.
Bariatric surgery is just the beginning to treat obesity. Commitment with the treatment especially with diet is capital for the success of this surgery.
I have some suggestions and answers that I would like to be answered as minor revisions:
Abstract
Line 16 - Please replace “on average form” by “on average from”
Line 22 - Please replace “natural nutrition” by “natural foods”
Line 22 - Please replace “is not sufficient” by “was not sufficient”
Line 22 - Please replace “The assumption of” by “The intake of”
Introduction
Line 39 - Please replace “that are assumed” by “that are consumed”
Line 59 - It lacks “1” before month
Line 60 - Please replace “natural nutrition” by “natural foods”
Materials and Methods
Line 119 - Please replace “drinks assumed” by “drinks consumed”
Line 124 - Please replace “dietician” by “dietitian”
Results
Table 1 - Title: Replace “before and 1 and 3 months after LSG” by “before and at 1 and 3 months after LSG”
Table 1 - Centre the ± (mean ± S.D.) and uniformize that in all table
Table 1 - Caption: Add the abbreviation of BMI (body mass index)
Line 182 - Add “s” to the word “evaluation”
Line 185 and 186 - Replace “additional intakes assumed through modular” by “additional intakes from modular”
Table 3 - Caption: Paired Student’s t test or Paired Student’s t test???
Line 193 - Replace “regularly assumed” by “regularly consumed”
Line 197 - Replace “who regularly assumed” by “who regularly consumed”
Lines 208 and 210 - Replace “meet” by “met”
Discussion
Lines 224 - Replace “regularly assuming the supplementation” by “regularly taking the supplementation”
Line 241 - “very low figures” - What does it mean?
Line 251 - Replace “in the assumption of solid aliments” by “in the consumption of solid aliments”
Line 253 - Add “s” to the word “consideration”
Line 256 - Replace “can be either assumed” by “can be either consumed”
Line 259 - Remove “and” from “surgery and”
Line 262 - Remove the space “of patients”
Line 262 - Replace “regularly assuming” by “regularly taking”
Round 2
Reviewer 1 Report
Congratulations on a paper that has dramatically improved! The article reads much better, has more coherent flow and covers an area where there has been little to no investigation and/or publications. I do have a few edits recommended before publlcation:
It is imperative that you put your sample size (n=x) for each timepoint into tables 1-4.
What was your drop-out rate? If none, please include this in the beginning of your results.
Line 51 your cite withinthe article (Mechanick et al., 2013) does not match your reference (Mechanick eta l., 2020). Please use the more recent -2020 reference in both places: article and references.
Please address the following comments in lines:
215 change regularly to regular
219 delete "of"
226 "The use of protein supplementation provides consents a significant significance increase in protein intake
228 "however" used incorrectly; use "further" or other terms instead of "however"
233 This proportion could be further increased in cases case of very rapid weight loss, like in particular during in the first ( Use "few" or "three") months after a bariatric procedure.
250-256 Please have someone familiar with the English language help you convey what you want to say in this paragraph. It is poorly written and "as is" should not be published.
Overall, you have a much better paper. Please incorporate the above edits. I look forward to seeing your work published soon.
Author Response
Reviewer #1
Congratulations on a paper that has dramatically improved! The article reads much better, has more coherent flow and covers an area where there has been little to no investigation and/or publications.
We thank the reviewer for these positive comments.
I do have a few edits recommended before publlcation:
It is imperative that you put your sample size (n=x) for each timepoint into tables 1-4. What was your drop-out rate? If none, please include this in the beginning of your results.
The sample size for each timepoint has been added into Tables 1-4. Our drop-out rate was none. This is now specified at the beginning of the results.
Line 51 your cite within the article (Mechanick et al., 2013) does not match your reference (Mechanick eta l., 2020). Please use the more recent -2020 reference in both places: article and references.
Thanks for noting. We changed the text from ”2013 Update” to “2019 Update” (published in 2020) in line 51.
Please address the following comments in lines:
215 change regularly to regular
Thanks for noting. We change “regularly” to “regular” in Line 217 of the revised manuscript.
219 delete "of"
Thanks for noting. We delete “of” in Line 219 of the revised manuscript.
226 "The use of protein supplementation provides consents a significant significance increase in protein intake
Thanks for the suggestion. We use “provides” in Line 228 of the revised manuscript.
228 "however" used incorrectly; use "further" or other terms instead of "however"
Thanks for the suggestion. We use “furthermore” instead of “however” in Line 228 of the revised manuscript.
233 This proportion could be further increased in cases case of very rapid weight loss, like in particular during in the first ( Use "few" or "three") months after a bariatric procedure.
Thanks for the suggestion. We use “like in particular during in the first few months” in Line 235 of the revised manuscript.
250-256 Please have someone familiar with the English language help you convey what you want to say in this paragraph. It is poorly written and "as is" should not be published.
We have some difficult with this paragraph. We try to simplify our message in this way: “The main reason for not achieving adequate protein intake after bariatric surgery is difficulty to consume solid protein-based aliments [23]. After SG, the consumption of solid protein-based foods (meat in particular) requires a careful and slow mastication and these foods can elicit more frequently nausea, vomit, and regurgitation of food. Finally, proteins are considered the macronutrient which satiates the most [24,25]”.
Overall, you have a much better paper. Please incorporate the above edits. I look forward to seeing your work published soon.
Thanks for your help. We are really grateful for your suggestion and we look forward to publication.
